# SoftTiger: A Clinical Foundation Model for Healthcare Workflows

**Ye Chen**[1*], **Igor Couto**[2*], **Wei Cai** [1*], **Cong Fu**[1], **Bruno Dornelles**[2]

[1]Tiger Research
Shanghai, China
{wei.cai, yechen, cong.fu}@tigerbot.com
[2]Sofya
Miami, Florida, USA
{igorcouto, brunodorneles}@sofya.ai

## Abstract

We introduce SoftTiger, a clinical large language model (CLaM) designed as a foundation model for healthcare workflows. The narrative and unstructured nature of clinical notes is a major obstacle for healthcare intelligentization. We address a critical problem of structuring clinical notes into clinical data, according to international interoperability standards. We collect and annotate data for three subtasks, namely, international patient summary, clinical impression and medical encounter. We then supervised fine-tuned a state-of-the-art LLM using public and credentialed clinical data. The training is orchestrated in a way that the target model can first support basic clinical tasks such as abbreviation expansion and temporal information extraction, and then learn to perform more complex downstream clinical tasks. Moreover, we address several modeling challenges in the healthcare context, e.g., extra long context window. Our blind pairwise evaluation shows that SoftTiger outperforms other popular open-source models and GPT-3.5, comparable to Gemini-pro, with a mild gap from GPT-4. We believe that LLMs may become a stepstone towards healthcare digitalization and democratization. Therefore, we publicly release SoftTiger models at scales of 13 billion and 70 billion parameters [1], as well as datasets and code for our innovative scalable evaluation, hopefully, making a significant contribution to the healthcare industry.

## Introduction

The healthcare sector is currently grappling with an unprecedented level of demand and critical challenges. In an ideal setting, physicians would require an unfeasible 26 hours per day to adhere to all care protocols, underscoring their extreme work pressure (Porter et al. 2022). This scenario is compounded by the fact of nearly half of physician's time is devoted to digital paperwork, rather than direct patient care (Sinsky et al. 2016). The intensive workload led to a disturbing increasing trend: 50.4% of physicians reported burnout, in a large cohort and longitudinal study (Ortega et al. 2023) with many specialities. Moreover, this overburden of healthcare professionals has dire consequences

---

*These authors contributed equally.

[1]models: https://huggingface.co/TigerResearch; data and code: https://physionet.org/projects/bWnuYbH3hewAnOONalfV

for patient safety. Nearly 800,000 patients annually in the United States (US) are harmed by diagnostic errors. Most of them were associated with cognitive mistakes, according to a recent study of John Hopkins (Newman-Toker et al. 2024), escalating medical errors as the third leading cause of death in the US (Makary and Daniel 2016). This scenario is already dramatic by itself, but it would even be worsened by the predicted shortfall of health workforce (HWF) of 18 million health workers by 2030 (Boniol et al. 2022), making a critical need for systemic changes in healthcare industry.

Recent advancements in large language models (LLMs), both proprietary models, e.g. GPT (Brown et al. 2020) and Gemini (Pichai 2023), and open-source models, e.g. Llama-2 (Touvron et al. 2023) and TigerBot (Chen et al. 2023a), have shown significant potential in processing and analyzing clinical notes (Kweon et al. 2023; Chen et al. 2023b). However, integrating them into clinical practice poses two primary challenges. The first pertains to the helpfulness of the AI powered clinical tasks, such as clinical note question answering (Kweon et al. 2023). These tasks, while functionally important, must also be designed and implemented seamlessly with both electronic health records (EHRs) and the clinical steps of the patient care to avoid workflow fragmentation (Moy et al. 2023). To understand the potential for enhancing the workflows, we conducted a survey of various clinical downstream tasks, from associated research papers from arXiv and AI models in Hugging Face, including clinical language models (CLaMs) and foundation models for electronic medical records (FEMRs) (Wornow et al. 2023). We asked domain experts to provide rate for a Fibonnaci-scale on the complexity and helpfulness of reported tasks, with results as shown in the Figure 1.

The second challenge is associated with the input length constraints of LLMs. Most popular LLMs were trained using 4k-token context window, e.g., Llama-2, which is sufficient for most general-purpose tasks. An annotated sample of 620 clinical notes drawn from MIMIC IV (Johnson et al. 2023) shows that 75% exceeds 2k and 38% is above 4k tokens, a Gaussian-like distribution very different from other daily tasks (usually follows a power law). Therefore, it is imperative for clinical LLMs to be able to efficiently trained on a context window up to 8k tokens or more, instead of length extrapolation during inference, which we found in our experiments would lose resolution drastically.

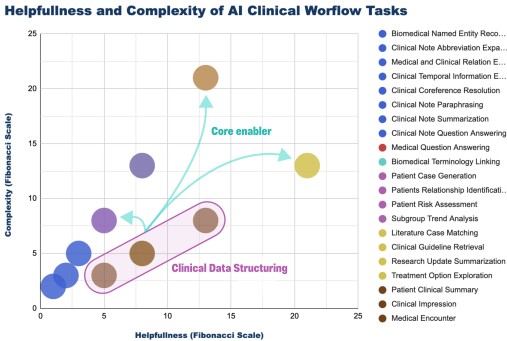

Figure 1: A survey on complexity and helpfulness of AI clinical tasks.

In this work, we strategically focused on the tasks of patient clinical data structuring, a crucial yet intricate component of clinical workflows. A large published review shows that almost 80% of medical data is unstructured (A.Maria Nancy 2020), hampering efficient development of more intelligent models. We choose these tasks, both moderately complex and highly valuable in our comprehensive analysis of the clinical domain, not only due to its practical significance, but also considering the status quo of LLMs, particularly their superb capabilities exhibited in structural extraction and natural language instruction following. We present SoftTiger, a clinical LLM suitable for both basic tasks such as named entity recognition, summarization and question answering, but also for more complex and foundational workflows tasks of clinical data structuring.

Our approach is seamless and lightweight embarking from a state-of-the-art (SOTA) general-purpose LLM, taking about 20 thousand dollars of GPU hours and a week to develop. The compute economics we deem is critical for rapid experimentation and adoption in the healthcare domain. Overall, we have made the following contributions:

- We publicly release a family of clinical LLMs, SoftTiger, at scales of 13 billion and 70 billion parameters, achieving SOTA performance in clinical note processing compared with other popular open and closed-source LLMs.

- We develop a stack of algorithmic and infrastructural implementations, not only fast adapting LLMs to the clinical domain, but also addressing challenges specific to the domain, including training long context and medical jargon understanding and abbreviation expansion.

- We also open-source release our first training data for the enablement of clinical workflow tasks, in a subset of 100 clinical notes sourced from MIMIC-IV free text notes dataset (Johnson et al. 2023), structured by Azure OpenAI GPT-4 and further curated and corrected by a team of experienced physician annotators.

- Furthermore, to validate the efficacy of SoftTiger in the light of fast experimentation and adoption, we implemented and also open-source a comprehensive LLM-as-a-Judge (Zheng et al. 2023) evaluation test of helpfulness and harmfulness of clinical data structuring.

## Problem Formulation

In order to build a clinical foundational model, our approach was not only focused into building the LLM capacities, but also the enablement of global digital health standards like the International Patient Summary (IPS) and HL7 Fast Healthcare Interoperability Resources (FHIR), to ensure robust and universally applicable solutions. In this first release, we aligned and optimized the model for three distinct subtasks, each focusing on a different aspect of patient information and interaction with the healthcare system. The aim is to facilitate better healthcare planning and efficient patient care through comprehensive and organized clinical documentation. The three subtasks are as follows:

1. **Patient Clinical Summary (FHIR IPS)** [2]: This subtask involves creating a comprehensive summary of the patient's social and clinical history. It includes detailing the patient's background, lifestyle choices, past illnesses, and family medical history. The objective is to provide a complete overview of the patient's medical and personal background, which is crucial for informed healthcare planning and decision-making.

2. **Clinical Impression (FHIR Clinical Impression)** [3]: The focus of this subtask is to summarize objective information gathered from various patient examinations. This includes documenting findings from imaging studies, laboratory test results, and other diagnostic procedures. The goal is to efficiently compile and review the patient's diagnostic data, aiding in the formulation of accurate clinical impressions and treatment plans.

3. **Medical Encounter (FHIR Encounter)** [4]: This subtask aims to systematically document the key elements of each patient-physician interaction. Essential details such as the location of the encounter, participants involved, and the reasons for the encounter are to be recorded. The purpose is to streamline the clinical documentation process, ensuring a clear and concise record of patient visits and the medical team's involvement.

By completing these subtasks, healthcare professionals can ensure thorough and efficient clinical documentation for every step, which is essential for quality patient care and effective healthcare management.

## SoftTiger Models

We are open-source releasing the SoftTiger family of clinical LLMs for free research and experimentation use, as summarized in Table 1. Figure 2 shows the training and validation loss for fine-tuning.

### Training Methods

SoftTiger models are supervised fine-tuned (SFT) on general-purpose open-source foundation LLMs, e.g., Llama-2 and TigerBot. Our choice of the foundation model is based upon several design considerations. First,

---

[2] Web: https://build.fhir.org/ig/HL7/fhir-ips

[3] Web: https://build.fhir.org/clinicalimpression.html

[4] Web: https://build.fhir.org/encounter.html

| Model | Chat | GPU hours | Train seqlen |
|-------|------|-----------|--------------|
| 13B | ✓ | 173 | 8k |
| 70B | ✓ | 416 | 8k |

Table 1: SoftTiger model family

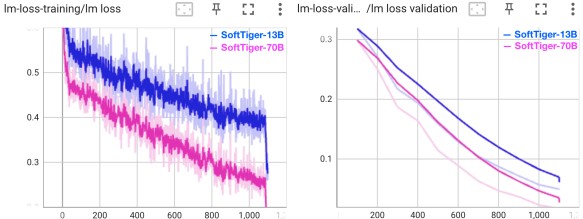

Figure 2: Training and validation loss for SoftTiger

the model has a good representativeness of biomedical vocabulary. Llama-2 has a vocabulary size of 30k, while TigerBot has 65k vocabulary. TigerBot is more transparent in open-source its training data (Chen et al. 2023a), which particularly includes the whole corpora of arXiv papers, besides academic books and Wikipedia. The arXiv dataset has 1.2% biomedical related subjects, which we deem lays us a good foundation knowledge for healthcare domain. Second, the foundation model should have grasped general-purpose tasks such as summarization, extraction, and question answering, with good instruction-following capabilities. For this, we choose chat models instead of pre-trained ones. Building a clinical LLM is essentially a domain adaptation process, which should be lightweight. Also, the clinical domain data is mostly one order of magnitude smaller than general fine-tuning data. Third, it is beneficial for world-wide adoption to build multilingual models with a range of parameter sizes. Llama-2 is known mainly for English, while TigerBot is a multilingual foundation model. Both model families have sizes from 7B, 13B, to 70B, while TigerBot further supports 180B.

We embark on TigerBot models [5] for our clinical LLM exploration. To empirically validate our choice, we collected a sample of 100 curated clinical notes from MIMIC IV, and then used Azure OpenAI GPT-4 to simulate three subtasks of patient clinical data structuring, to form a validation dataset of 300 examples. We then evaluate TigerBot and Llama-2 chat models using next-token prediction. The results support our choice of TigerBot, as shown in Table 2.

| Model | Eval seqlen | Accuracy |
|-------|-------------|----------|
| Llama-2-70b-chat | 4k | 0.8155 |
| | 8k | 0.405 |
| TigerBot-70b-chat | 4k | 0.8182 |
| | 8k | 0.8743 |

Table 2: Foundation model evaluation

[5] web: https://www.tigerbot.com/chat; github: https://github.com/TigerResearch/TigerBot

Both foundation models were trained on 4k context length, without domain data fine-tuning. For both 4k and 8k evaluation context windows, TigerBot outperforms Llama-2. One thing noteworthy, when evaluated using 8k context length, Llama-2 performs catastrophically worse. We conjecture that this is due to under representativeness of clinical vocabulary which leads to worsened hallucination.

**Training Data**

Our training data consists of 134 million tokens, or 313k instruction completion examples and 489M plain-text data in the JSON format: {"instruction":..., "output":...}. For the clinical domain data, the "instruction" prompt is composed of input clinical note concatenated with task instruction, while the "output" is structural extraction. Input clinical notes were drawn from MIMIC IV dataset. The three output task extractions were first synthesized from Azure OpenAI GPT-4, and then corrected for quality and safety by a group of 5 physicians during one week. They received an annotation manual and two hour remote training in the task domain. Also, they followed an annotation workflow of to balance workload.

The histogram of data lengths in tokens is shown in Figure 3. Other than output length, both input and total length of examples follow a Gaussian-like distribution, with a major chunk beyond 2k. The use of medical term abbreviation is a norm in clinical notes. We employ a dictionary of abbreviation expansion to standardize abbreviations both for training and inference runtime.

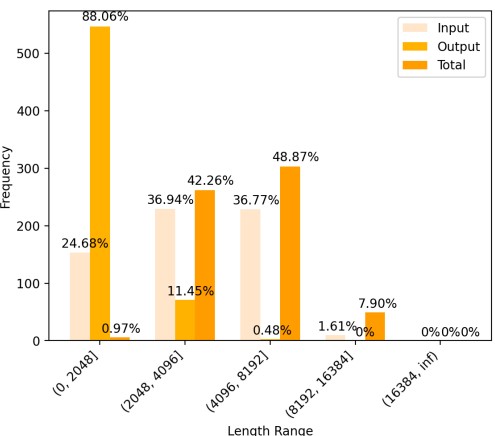

Figure 3: The distribution of training data length in tokens.

Since we formulate the clinical LLM building as a domain specialization problem, we mix in two layers of data other than the domain data for the task of patient summarization. First, a new sample of general-purpose SFT data was drawn from previously unseen corpus. This layer is for experience replay (Sun et al. 2020) for general tasks such as generation, summarization, question answering, and so forth. Second, we add the Asclepius dataset (Kweon et al. 2023) for basic functional tasks in the clinical domain, such as named entity recognition, abbreviation expansion, and paraphrasing, etc. The detailed training data mix is illustrated in Table 3.

We organize the training data in an order from general, domain, to task specific. The trainer scans the data sequentially within one epoch, following human learning process of domain specialization.

| Dataset | Tokens (M) | Size (M) |
|---|---|---|
| General purpose | 16.4 | 79.6 |
| Domain - Asclepius | 111 | 396 |
| Task - annotated | 6.5 | 13.6 |
| **Total** | **134** | **489** |

Table 3: SoftTiger training data mix

## Training Framework

SoftTiger models have been trained using our proprietary codebase enhanced from Megatron-DeepSpeed (Chen et al. 2023a; Microsoft 2023). Tensor parallelism (TP) is particularly critical since model size over 70B cannot fit into a single GPU while CPU offloading is slow and we want to avoid. 3D parallelism gives a maximal optimization space for speed and memory tradeoff, under different scenarios and resource settings. In our setting of clinical notes, training on long sequence is a prerequisite, given the data distribution shown in Figure 3. Our training cluster consists of $64\times$ A100-40G GPUs. After a quick geometric search, we set $TP = 8$, $PP = 8$ and $DP = 1$, to minimize inter-node intra-tensor communications while preserving most GPU memory for 8k sequence length.

## Evaluation and Alignment

We took a two-stage evaluation approach. First, we used an evaluation dataset of 300 examples, with 100 clinical notes sampled from MIMIC IV and three task completions for each executed by Azure OpenAI GPT-4. We then perform next-token prediction programmatically, which is fast and critical for rapid iteration. The automated evaluation results are shown in Table 4. SoftTiger models outperform Llama-2 for both model sizes and evaluation context lengths. SoftTiger-70b model surpasses 13b by a small margin, possibly because the task-specific training data is still small in volume and the learning is still on the monotonically increasing trajectory.

| Model | Eval seqlen | Accuracy |
|---|---|---|
| Llama-2-13b-chat | 4k | 0.8002 |
| | 8k | 0.3009 |
| Llama-2-70b-chat | 4k | 0.8155 |
| | 8k | 0.405 |
| SoftTiger-13b | 4k | 0.8694 |
| | 8k | 0.8724 |
| SoftTiger–70b | 4k | **0.8707** |
| | 8k | **0.8784** |

Table 4: Next-token prediction results

Secondly, we implemented a ChatBot Arena with LLM-as-a-Judge using Azure OpenAI GPT-4. We followed blind pairwise evaluation method (ModelA vs ModelB), with swap-position executions to prevent positional bias (changed positions with different results are discarded). Study shows average of 85% of agreement with humans using GPT-4 (Zheng et al. 2023). The arena dataset consists of 100 answers for the Patient Clinical Summary task by popular both closed and open-source LLM models. We also designed a control group model, with intentionally wrong information. We determined the intelligence inspired chess ELO rating [6] (initial rating=1000, k=30) as in Table 5, with some illustrations of SoftTiger and implementation details of the blind test shown in Appendices.

| Model | ELO | Wins | Loses | Battles |
|---|---|---|---|---|
| GPT-4-0125-preview | 1387 | 182 | 62 | 244 |
| Gemini-pro | 1167 | 132 | 131 | 263 |
| SoftTiger-70b | 1142 | 118 | 126 | 244 |
| SoftTiger-13b | 1013 | 90 | 141 | 231 |
| GPT-3.5-turbo-1106 | 960 | 96 | 166 | 262 |
| Mixtral-8x7b | 937 | 77 | 185 | 262 |
| Control group | 390 | 0 | 253 | 253 |

Table 5: ELO ratings for Patient Clinical Summary task

This cost-effective evaluation method rapidly enables iterative experimentation and optimization at scale, allowing further optimized human alignment with healthcare professionals. Reasonable results of bigger closed models as higher ELO, but optimized SoftTiger-70b near Gemini-pro. Lightweight and cost-effective SoftTiger-13b model also showing feasibility for deployment and surpassing GPT-3.5-Turbo and Mixtral-8x7b.

## Conclusions

In this work, we have developed the SoftTiger clinical LLMs to tackle the foundational problem of patient clinical data structuring. The task is very practical and valuable to the clinical workflows, and hopefully becomes our first step towards deeper patient care insights and intelligence in future tasks. Usually physicians deal with high dense and noisy narrative data. Giving them this level of structure and summary helps to alleviate the cognitive pressure and the dramatic situation of healthcare delivery overburden.

We leveraged LLM's excellent capabilities of structural extraction and natural language interaction to achieve satisfactory results, specially in comparison with other popular yet general models, using innovative rapid experimentation and evaluation methods . However, the statistical nature of LLMs still incur hallucination issues, which is particularly risky in medical domain and demand more work towards model alignment. Our future works plan to tackle the hallucination problem using retrieval-augmented generation (RAG) with expert-curated knowledge graphs with biomedical terminologies. We also plan to explore specially designed reinforcement learning algorithms to increase model intelligence and execute the first real cases of clinical workflows integration experiments.

---

[6] Web: https://www.chess.com/terms/elo-rating-chessratings

**Appendix A: Clinical Note Sample**

Pt is a ___ y/o R-handed male w/ PMHx of GERD, +PPD s/p INH, depression, peripheral vertigo, & shrapnel in L hand/leg, presenting w/ acute onset diplopia & dizziness. No significant surgical Hx. Allergic to Protonix. Neuro exam notable for R abducens nerve palsy; other CNs intact. Motor/sensory/DTRs WNL. CT head showed questionable hypodensity in midbrain; no MRI due to shrapnel. Labs show mild CRP elevation, normal ESR, A1C, and cholesterol levels. Lyme serology neg. CTA head/neck unremarkable. D/c w/ dx of likely peripheral abducens nerve palsy. Pt alert & ambulatory at d/c. D/c meds: omeprazole & ASA 81 mg PO daily. FU w/ PCP & stroke neurology. Counseled on expected gradual improvement of diplopia, continue ASA, and importance of FU for reassessment.

Pt stable for d/c to home. Educated on likely peripheral etiology of diplopia, expectations for improvement, & importance of ASA therapy. Urged to maintain FU appts for ongoing eval & mgmt.

Figure 4: Sample MIMIC-IV clinical note synthetically summarized

---

**Appendix B: Examples of SoftTiger at work - Patient Clinical Summary**

**Header Information**
- **Subject:** Male

**Medication Summary**
- **Name:** Omeprazole
  - **Dosage:** Not specified
  - **Route:** Not specified
  - **Frequency:** Daily
  - **Period:** Not specified
- **Name:** Acetylsalicylic Acid (ASA)
  - **Dosage:** 81 mg
  - **Route:** Not specified
  - **Frequency:** Daily
  - **Period:** Not specified

**Allergies and Intolerances**
- **Allergen:** Pantoprazole (Protonix)

**Problem List**
- **Condition:** Gastroesophageal Reflux Disease (GERD)
  - **Clinical Status:** Not specified
  - **Onset:** Not specified
- **Condition:** Peripheral Vertigo
  - **Clinical Status:** Not specified
  - **Onset:** Not specified
- **Condition:** Depression
  - **Clinical Status:** Not specified
  - **Onset:** Not specified
- **Condition:** Right Abducens Nerve Palsy
  - **Clinical Status:** Not specified
  - **Onset:** Not specified

Figure 5: Patient Clinical Summary Example from Clinical Note - to be continued

# Appendix B: Examples of SoftTiger at work - Patient Clinical Summary - continued

## Problem List - continued

- **Condition:** Shrapnel in Left Hand/Leg
  - **Clinical Status:** Not specified
  - **Onset:** Not specified
- **Condition:** Diplopia
  - **Clinical Status:** Not specified
  - **Onset:** Acute
- **Condition:** Dizziness
  - **Clinical Status:** Not specified
  - **Onset:** Acute

## Social History

- **Factor:** Not specified

## History of Procedures

- **Procedure Name:** Not specified
- **Procedure Date:** Not specified
- **Body Site:** Not specified

## Medical Devices

- **Device Name:** Not specified

## Diagnostic Results

- **Test Name:** Computed Tomography (CT) Scan of the Head
  - **Result:** Questionable hypodensity in the midbrain
- **Test Name:** Post-Infectious Hydrocephalus (INH) Test
  - **Result:** Positive
- **Test Name:** Magnetic Resonance Imaging (MRI)
  - **Result:** Not specified due to shrapnel
- **Test Name:** Computed Tomography Angiography (CTA) of the Head and Neck
  - **Result:** Unremarkable
- **Test Name:** Laboratory Tests
  - **Result:** Mild C-reactive protein (CRP) elevation, normal erythrocyte sedimentation rate (ESR), hemoglobin A1C, and cholesterol levels

## Vital Signs

- **Observation:** Not specified
- **Value:** Not specified

## Plan of Care

- **Care Plan Description:** Discharged with diagnosis of likely peripheral abducens nerve palsy, advised to follow up with primary care physician and stroke neurologist, counseled on expected gradual improvement of diplopia, continue ASA, and importance of follow-up for reassessment

Figure 6: Patient Clinical Summary Example from Clinical Note - continued

# Appendix C: Examples of SoftTiger at work - Patient Clinical Impression

## Basic Information

- **Status:** Completed
- **Status Reason:** Patient discharged with follow-up plan
- **Description:** Assessment of male patient with acute onset diplopia and dizziness

## Subject and Encounter Details

- **Subject:** Male patient, age not specified
- **Encounter:** Presentation with acute onset diplopia and dizziness

## Assessment Timing

- **Effective DateTime:** Not specified
- **Effective Period:** Not specified
- **Date of Assessment Documentation:** Not specified

## Assessment Performer

- **Performer:** Attending physician, Not specified

## Clinical Context

- **Previous Assessment:** Not specified
- **Problems/Conditions:** Gastroesophageal reflux disease (GERD), Positive PPD test, Depression, Peripheral vertigo, Shrapnel in left hand/leg
- **Change Pattern:** Acute onset diplopia and dizziness
- **Protocol Followed:** Not specified

## Summary and Findings

- **Summary:** Male patient presented with acute onset diplopia and dizziness. No major surgical history. Allergies to Pantoprazole (Protonix). Neuro exam revealed right abducens nerve palsy. CT head showed questionable hypodensity in midbrain. No MRI due to shrapnel. Labs showed mild CRP elevation, normal ESR, A1C, and cholesterol levels. Lyme serology negative. CTA head/neck unremarkable. Discharged with likely peripheral abducens nerve palsy.
- **Findings:**
  - Right abducens nerve palsy (Basis: Neuro exam)
  - Hypodensity in midbrain (Basis: CT head)

## Prognosis

- **Prognosis Codeable Concept:** Not specified
- **Prognosis Reference:** Not specified

## Supporting Information

- **Supporting Info:** Past medical history of GERD, positive PPD test, depression, peripheral vertigo, and shrapnel in left hand/leg. Allergies to Pantoprazole (Protonix). Neuro exam findings. CT head and CTA head/neck imaging results. Labs results showing mild CRP elevation, normal ESR, A1C, and cholesterol levels. Discharge medications: omeprazole and ASA 81 mg Oral (PO) daily.

## Notes and Comments

- Patient advised on likely peripheral etiology of diplopia
- Expected gradual improvement of diplopia
- Continue ASA therapy
- Importance of follow-up appointments for reassessment

Figure 7: Patient Clinical Impression Example from Clinical Note

# Appendix D: Examples of SoftTiger at work - Encounter Summary

**Basic Information**

- **Status:** Completed
- **Class:** Outpatient

**Encounter Details**

- **Priority:** Non-urgent
- **Type:** Consultation
- **Service Type:** Neurology

**Subject Information**

- **Subject:** Not specified
- **Subject Status:** Departed

**Contextual Links**

- **Episode Of Care:** Not specified
- **Based On:** Not specified
- **Care Team:** Not specified
- **Service Provider:** Not specified

**Timing Information**

- **Actual Period:** Not specified
- **Planned Start Date:** Not specified
- **Planned End Date:** Not specified
- **Length:** Not specified

**Reasons and Diagnoses**

- **Reason:** Chief Complaint - Diplopia, dizziness
- **Diagnosis:** Peripheral abducens nerve palsy (Use: Admission)

**Billing and Preferences**

- **Diet Preference:** Not specified
- **Special Arrangement:** Not specified
- **Special Courtesy:** Not specified

**Admission Details**

- **Admission Origin:** Not specified
- **Admit Source:** Not specified
- **ReAdmission:** Not specified
- **Destination:** Not specified
- **Discharge Disposition:** Home

**Location Information**

- **Location:** Emergency department
- **Status:** Completed
- **Form:** Not specified
- **Period:** Not specified

Figure 8: Patient Medical Encounter Interaction Example

## Appendix E: LLM-as-judge Evaluation Prompt

**Evaluation Overview**

Act as an impartial judge to evaluate the performance of two AI models, Model A and Model B, in processing and structuring data from clinical notes. Focus on helpfulness, harmlessness, relevance, and accuracy.

**Structuring Information**

- Assess the proficiency in organizing information into appropriate sections: Header Information, Medications, Allergies and Intolerances, Problem, Social History, History of Procedures, Medical Devices, Diagnostic Results, Vital Signs, Plan of Care.
- Each term in the section must be structured as a single item. Penalize for cases not itemized.

**Information Reliability**

- Evaluate adherence to the original clinical note.
- Penalize for fabrication or invention of data.
- Data not in the original note and inserted by the model is considered fabrication.

**Extraction and Relation of Medical Attributes**

- Judge accuracy in extracting correct information.
- Evaluate how well information is correlated with its respective attributes in the clinical note.

**Scoring and Decision**

- The winning model is the one with the highest total score, considering all criteria. Data fabrication is a is a decisive criteria for evaluation and is also a tiebreaker.

Figure 9: LLM-as-judge Evaluation Prompt

## Appendix F: Sample LLM-as-judge Evaluation Result

**Evaluation Summary**

**Winner:** Model B

**Detailed Explanation**

- Model B excelled in structuring information into appropriate sections with high detail and specificity, especially in the Problem List.
- It adheres closely to the original clinical note, ensuring no data fabrication.
- Model B accurately extracts and correlates medical attributes like conditions, treatments, and diagnostic results with their details.
- Despite both models needing improvement in specifying medication details, Model B's approach in detailing patient conditions and treatments is superior.
- Notable strengths include the addition of oral route for warfarin and specific conditions like fevers, night sweats, and back pain.
- Based on these factors, Model B is awarded the highest total score across all evaluation criteria.

Figure 10: AI Model Evaluation Result

## Ethical Considerations and Reproducibility Statement

In the development and deployment of SoftTiger, a clinical large language model (CLaM), we have rigorously adhered to ethical guidelines and best practices to ensure the responsible use of technology in healthcare. We recognize the sensitive nature of clinical data and the critical importance of maintaining patient privacy and safety. This statement outlines the ethical considerations we have taken into account and the steps we have taken to ensure the reproducibility of our research and work.

- **Data Privacy and Confidentiality:** All patient data used in training SoftTiger were sampled from the Medical Information Mart for Intensive Care (MIMIC-IV) dataset, already complying with patient data de-identification and safety cleaning.

- **Data Responsible Use:** According to PhysioNet Credentialed Health Data License 1.5.0 [7], and Physionet Responsible use of MIMIC data with online services like GPT [8], Azure OPENAI Service and AWS Bedrock were used to process data. The data and code of this work is also available under Physionet terms and repository.

- **Bias and Fairness:** We acknowledge the potential for inherent biases in AI models. To mitigate this, in the evaluation method of ChatBot-Arena we took the conservative approach of repeating the evaluations twice while switching positions and only considering agreed results. The evaluation was blind sided, also with a control group to detect deviations from judgment. We also randomly sampled data in large MIMIC datasets considering mixed sized and complexity distributions in inputs.

- **Transparency and Openness:** In line with promoting reproducibility, we have made SoftTiger's datasets, training data, and model parameters publicly available. All the prompts used and results, including all of the code necessary to generate, expand and recreate the datasets are open-source. All the prompts, data and code to reproduce the ChatbotArena evaluation are also available.

- **Safety and Reliability:** A core design principle of SoftTiger responsible AI development.For this, we included helpfulness and harmfulness of the models as a core criteria of every evaluation, and warn everyone using our work to further align any real world clinical use first with a human-eval of healthcare professionals considered as a last step of model alignment, then with a Ethics Board Approval of the healthcare institution.

- **Ongoing Monitoring and Improvement:** Recognizing that AI in healthcare is an evolving field, we commit to continuous monitoring of SoftTiger's performance and making iterative improvements to address emerging issues or changes in clinical practices.

- **Compliance with Regulatory Standards:** Our development process and model deployment are in compliance with relevant healthcare regulations and standards, ensuring that SoftTiger aligns with legal and ethical requirements. This was particularly considered in the format of the clinical downstream tasks, by selecting the International Patient Summary (IPS) and HL7 FHIR entities (Clinical Impression, Medical Encounter)

- **Collaboration with Healthcare Professionals:** Throughout development and deployment, we have actively collaborated with healthcare professionals to ensure that SoftTiger meets the practical needs of the healthcare industry and aligns with clinical workflows.

## Ethics Board Approval

For this initial release of SoftTiger models, involving the development and training of the clinical large language model

(CLaM), an Ethics Board approval was not deemed necessary. This decision is grounded in the fact that our work primarily involved the use of de-identified, publicly or credentialed available data and did not directly engage with human subjects or patient-specific data in a clinical setting.

However, we strongly advise that any derivative works, extensions, or real-world clinical implementations of Soft-Tiger undergo thorough ethical review and obtain necessary approvals. This is particularly crucial when such projects involve:

- The use of personally identifiable patient data or engagement with human subjects.

- Implementation in clinical settings where decisions may directly affect patient care and outcomes.

- Any form of clinical trial or research that involves human participants.

We emphasize the importance of responsible clinical alignment, ensuring that any use of SoftTiger or its derivatives aligns with ethical standards, respects patient privacy and confidentiality, and adheres to all applicable regulations and guidelines. This approach is vital for maintaining public trust and ensuring the responsible use of AI in healthcare.

## Limitations and Future Work

As the first LLM fine-tuning focused in clinical data structuring according to healthcare interoperability guidelines, this work has some limitations that we expect to overcome in future versions and following articles. These were:

- **Lack of measures in data curation and annotation for model training::** In future versions, we expect to evolve data governance transparency in terms of diversity of selection of clinical notes, inter-agreement and recall of the pre-structured annotation data from Azure OpenAI GPT-4 and experts as Kappa/Cohen and also ROUGE, as well as charts about data distribution.

- **Usage of only clinical discharges in training and evaluation data**: As we used only MIMIC-IV clinical discharges to train and evaluate the model performance, we do expect some variations in performance in other scenarios, like social medial narratives or ambulatory clinical notes. Next versions should address more diversity in training data as well as multi-language.

- **Stability of FHIR entity mappings**: As an early exploration work, we do expect minor adjustments in FHIR mappings, as the LLM output in markdown can output also semi structure data. Also, additional work will be required to process the output and conform to all FHIR specifications, as well as patient identification.

- **No concept coding**: No terminology coding techniques were applied. This can be achieved by iterating structured sections and calling Clinical Entity Taggers and Terminology linkers available, open-source or commercial.

- **Lack of proof on agreement between LLM-as-Judge and experts in domain**: Due to time constraints, no inter-agreement was calculated for the specific problem domain between LLM-as-judge and domain experts. We

---
[7]Web: https://physionet.org/about/licenses/physionet-credentialed-health-data-license-150/

[8]Web: https://physionet.org/news/post/gpt-responsible-use

see high correlation (= 85%) in the literature for general domains, but in future work we expect to measure in the specific problem.

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
