# OpenReview forum: "SoftTiger: A Clinical Foundation Model for Healthcare Workflows"
_AAAI.org/2024/Spring_Symposium_Series/Clinical_FMs — AAAI 2024 SSS on Clinical FMs_

### Official Review · Reviewer_WXeH · 2024-02-19
**Outlines the finetuning of an open source LLM for clinical data structuring. However there are large methodolgical flaws and some base assumptions are unconvincing.**

**Rating:** 3
**Confidence:** 3

**Review:**

1. Summary and contributions: Briefly summarize the paper and its contributions
Outlines the development of an LLM called SoftTiger. This is a finetuned version of an open-source LLM named TigerBot. This is achieved using supervised finetuning tuning from a dataset of general text, a previously released clinical dataset and a novel clinical workflow dataset. The novel dataset is made up of instruction pairs for 3 tasks performed on the MIMIC-IV dataset with the outputs generated by GPT-4 and validated by 5 physicians. The results show that the finetuned models gained accuracy on automated evaluation benchmarks.

2. Strengths: Describe the strengths of the work. Typical criteria include: soundness of the claims (theoretical grounding, empirical evaluation), significance and novelty of the contribution, and relevance to the community.
- It is an early example of finetuning large (70B) open-source LLMs across multiple GPUs.
- After carefully evaluating the trade-off between clinical complexity and helpfulness, 3 clinical data structuring tasks were chosen. This gives the work a clear potential for clinical impact.
- A very good section outlines the administrative burden on physicians.
- Making IPS or FHIR structure the output is optimal for potential future integration into current e-health systems

3. Weaknesses: Explain the limitations of this work along the same axes as above.
-MIMIC data user agreement prevents the sharing of the data or derivates with 3rd parties. Therefore, the SoftTiger and dataset should not be publicly released. They could be hosted on PhysioNet though.
- Similarly, MIMIC data should not be sent to 3rd party LLM providers as seems to have occurred in Table 5 unless via Azure or Amazon (see https://physionet.org/news/post/gpt-responsible-use). Please state clearly in text or “Ethical Considerations and Reproducibility Statement” if these services were used.
- Evaluation and training data only uses MIMIC-IV, which are discharge summary notes from the ICU department of a single health centre. This should be noted as a limitation.
- GPT-4 is used to produce the clinical training and evaluation set. This is then corrected by clinical review. No mention of the performance of GPT-4 on the task is made or the inter-annotator agreement. Furthermore, justification (most likely from a data governance perspective) is given on why GPT-4 cannot be used for this task directly if it can produce the labels for the task.

4. Correctness: Are the claims and method correct? Is the empirical methodology correct?
- In the introduction, it is claimed the 2 primary challenges for LLM clinical adaptation are finding a ‘helpful clinical task’ and input length constraint. I do not believe this to be true. Numerous clinical tasks could be performed by LLMs, e.g. diagnoses, discharge summary writing, reporting of adverse drug events etc... Barriers such as effective and safe evaluation, data privacy and governance, and integration into healthcare providers’ electronic systems would seem equal if not greater to this constraint.
The second constraint of input length is notable but only for open-source models (closed-source models have context lengths >100k), a distinction that is not made. However, the trained model is only extended to 8k and claimed as a source of novelty. Current open-source models such as mistral have been trained with an 8k context window.
- Claimed that note length usually follows power law without proof or citation
- The approach is claimed to be “light-weight” but requires 64xA100s GPUs
- It is claimed that as TigerBot has a larger vocabulary size than Llama-2, it has a larger clinical vocabulary. However, as TigerBot is multilingual this claim only seems true if evaluating on multilingual data also. The claim that TigerBot has a greater English clinical vocabulary needs further explanation or proof.
- It is not obvious that the addition of the general-purpose or Asclepius datasets will improve performance on the clinical workflow tasks.
- “Dictionary of abbreviation expansion to standardize abbreviations” is known not to work due to the redundancy of terms. For example, “hr” could be expanded to hour or heart rate, depending on the context.

5. Clarity: Is the paper well written?
- “We then evaluate TigerBot and Llama-2 chat models using next-token prediction.” It is not clear to me how this evaluation works. Is this exact matching? Further explanation is required.
- Not clear how Llama-2 and TigerBot were extended from 4k to 8k inputs.
- It is claimed that “it is beneficial for worldwide adoption to build multilingual models”, which is true. But it is not clear if the fine-tuning dataset is also multilingual.
- Fig.1 shows some very helpful information, but it is not clear which task is related to which plot point due to the use of repeated colours. Furthermore, the Fibonacci scale is not explained.
- Not clear if, in normal practice, discharge summaries are the only source of information used to complete the 3 subtasks trained and evaluated in this work.
- Figure 2 is a direct screenshot from tensorboard or similar. Removal of the UI buttons, and adding full and axis titles would improve this figure. Not clear what the faint lines are in the figure
- Table 3 should be moved higher up the training data section and would be more instructive swap the size column for number of examples in each dataset.
- Figure 3’s final column is all 0% and so does not need to be included. Moreover, the information may be more helpfully presented as a table of min, median, max for input, output and total

6. Relation to prior work: Is it clearly discussed how this work differs from previous contributions?
- This is the first open-source LLM finetuning to output on FHIR IPS, FHIR Clinical Impression and FHIR Encounter from medical discharge summaries.

7. Reproducibility: Are there enough details to reproduce the major results of this work?
- The number, speciality, nationality, and seniority of clinicians surveyed to produce Fig 1 is not stated
- Would be useful to link or add in the appendices the exact FHIR structures of the 3 subtasks.
- The training framework section is limited. No training hyperparameters are given. The acronyms PP and DP are used without explanation.
- The settings, prompt, and model version used to generate the clinical workflow dataset using GPT-4 are not stated

---

### Official Review · Reviewer_VttH · 2024-02-23
**This study presents high-quality research in the clinical domain, addressing challenges in building Clinical Large Language Models (CLaMs). It demonstrates originality by introducing novel models for structuring patient clinical data, with clear organization and thorough evaluation methods. The significance lies in its contribution to addressing the crucial problem of conforming clinical notes to international interoperability standards, showcasing superior performance compared to existing models.**

**Rating:** 7
**Confidence:** 5

**Review:**

># Quality
The work is of high quality, as it demonstrates a thorough understanding of the clinical domain and the challenges of building a clinical large language model (CLaM). The authors justify their choice of foundation model clearly. They also address several modeling challenges, such as long context window, medical jargon, and abbreviation expansion. Their models are evaluated using both next-token prediction and blind pairwise comparison with other popular LLMs, showing superior performance in patient clinical data structuring.

># Clarity
The work is well-written and organized, with clear problem formulation, methods, results, and discussion. The authors provide sufficient details and explanations for their data collection, model training, and evaluation methods, including several appendices with examples of their models’ outputs, ethical considerations, and reproducibility statement.

># Originality
This work is indeed original as it introduces a novel family of CLaMs designed for patient clinical data structuring, a crucial yet intricate component of clinical workflows.

># Significance
This work is significant, as it addresses a critical problem of structuring clinical notes into clinical data, according to international interoperability standards.

># Pros
* This work tackles an important area of healthcare and has a lot of potential significance.
* The work evaluates the models using both next-token prediction and blind pairwise comparison with other LLMs

---

### Official Review · Reviewer_Acro · 2024-02-23
**While the innovative approach in addressing healthcare challenges is commendable, potential drawbacks, such as the risk of hallucination and dependency on training data, underscore the importance of recognizing limitations for a comprehensive evaluation.**

**Rating:** 7
**Confidence:** 3

**Review:**

The strengths of the paper include its emphasis on high quality and clarity, showcasing SoftTiger's superior performance compared to established models like GPT-3.5. The clear articulation of objectives and addressing challenges in healthcare workflows adds to the paper's credibility.
The originality and significance of SoftTiger's approach in tackling critical subtasks within healthcare are commendable, contributing to its innovative standing. The acknowledgment of both the advanced capabilities and scalability of SoftTiger, with two configurations catering to different research needs, adds to its appeal.

However, the potential challenges or cons associated with SoftTiger, such as the risk of hallucination due to the statistical nature of language models and the dependency on the volume and quality of training data. Recognizing these limitations is essential for a comprehensive evaluation.

---

### Official Review · Reviewer_JAru · 2024-02-23
**SoftTiger review**

**Rating:** 9
**Confidence:** 4

**Review:**

This ambitious paper fine-tuning a large SOTA LLM for structuring clinical notes capable of handling long context windows and rigorously evaluates it to other commercial/open models using a chatbot arena + LLM-as-judge setup.

The background is well set with both a survey of clinical LLM tasks from arXiv and huggingface and an empirical investigation of context windows required for clinical notes in MIMIC-IV.

They chose a TigerBot base model as it is multilingual and demonstrated superior performance to Llama 2-70b chat with accuracy of next token prediction on 8k contexts. Whilst next token prediction isn’t a particularly useful task they justify it as suitable for rapid decision making and early exploration. The poor performance of Llama 2 70b on 8k vs 4k tokens is hypothesised due to under representativeness of clinical vocabulary leading to worsened hallucination - this seems possible, but I can’t believe that the justification that TigerBot was trained on arXiv which has 1.2% biomedical content explains the difference, I’m sure this was part of Llama training too!

The training data is constructed using clinical notes from MIMIC-IV processed using GPT-4. It’s good to see (trained!) expert evaluation of the training data. The ethical provisioning for this needs to be mentioned as PhysioNet does not permit use of OpenAI APIs! This is not mentioned despite the extensive ethics appendix.

Given the importance of training data it would be useful to have some additional details of how GPT-4 was used to restructure data for the three extraction tasks. There are some additional training datasets such as a ‘previously unseen corpus’ of general-purpose SFT data which needs clarifying. The use of ASclepius for basic clinical tasks like NER and abbreviation seems sensible but despite training for abbreviations they use a medical dictionary at both training and inference?

They structure training from general purpose to basic tasks (ner/abbreviations) to hard tasks (summarisation) but it would be nice to see some experimental results or referenced justification for why they did this. Otherwise this really follows the general LLM -> domain specific fine-tuning paradigm so I’m not sure constitutes a novel strategy in and of itself?

The technical details of training framework are clear and impressive.

Evaluation is rigorous with comparison with range of commercial and open-source (GPT3.5/4, Gemini Pro, llama2, mixtral). The use of ChatBot Arena for blind pairwise evaluation includes control groups with intentionally wrong information and swap-position executions. They use GPT-4 as a judge, supplying the evaluation prompt which is thorough, however despite using 5 domain experts to review training data I can’t see any expert review of the final evaluation data or the LLM as judge strategy? This appears to me to be the biggest weakness in an otherwise strong paper and is perhaps time related?

Overall this is a really ambitious and thorough piece of work which makes both an interesting contribution to the research literature as well as the open source community through release of trained models, datasets and evaluation code.